# Relationships between Mortar Spread and the Fresh Properties of SCC Containing Local Metakaolin

Abderrazak Barkat [1], Said Kenai [2,*], Belkasem Menadi [2], El-Hadj Kadri [3] and Jamal Khatib [4,*]

1. Energies and Materials Research Laboratory, University of Tamanghasset, Tamanghasset 11001, Algeria; azbarkat@univ-tam.dz
2. Geomaterial Laboratory, Department of Civil Engineering, University of Blida 1, Blida 09000, Algeria; b_menadi@univ-blida.dz
3. Laboratoire de Mécanique et Matériaux du Génie Civil Laboratory, CY Cergy-Paris University, F9500 Cergy Pontoise, 95031 Neuville-sur-Oise, France; el-hadj.kadri@cyu.fr
4. Faculty of Engineering, Beirut Arab University, Beirut P.O. Box 11 5020, Lebanon
* Correspondence: s.kenai@univ-blida.dz (S.K.); j.khatib@bau.edu.lb (J.K.)

**Abstract:** Self-compacting concrete (SCC) production is a complex operation that requires finding a good combination and suitable dosages for its constituents. Several formulation methods have been developed to meet the workability requirements of SCC. Mortar spread is used to estimate SCC's rheological properties, but the use of supplementary cementitious materials, such as metakaolin, could affect the accuracy of the estimation. In this paper, the relationships between the fresh properties of local-metakaolin (MK)-based SCC and the spreading of its mortar portion were investigated. The results showed the existence of good correlations between the spreading of mortar portion of SCC and its fresh state properties. The partial substitution of cement with MK did not affect these correlations. The mortar flow should be chosen according to the required rheological properties of the SCC. This can be achieved by using an appropriate viscosity-enhancing agent (VEA).

**Keywords:** metakaolin; SCM; SCC; mortar spread; flow time; viscosity

## 1. Introduction

Self-compacting concrete (SCC) is distinguished from ordinary concrete mainly by its properties in the fresh state. SCC requires the use of more fine materials such as fine aggregate, cement replacement, and filler materials in order to reduce segregation or bleeding during transportation and placement [1,2]. The high flowability of SCC contributes to the reduction in casting time through the elimination of vibration and a reduction in noise pollution [3]. The design of an SCC mixture is based on the same criteria usually chosen for the formulation of vibrated concretes (i.e., workability, strength, and durability). In addition to these criteria, SCC must satisfy two contradictory properties in its fresh state: good fluidity to ensure good placement and good viscosity to guarantee adequate resistance to segregation.

Several SCC mix design methods have been developed, based on different guidelines, such as the Japanese, Swedish CBI, and French LCPC guidelines, in order to meet the workability requirements [4]. In this work, the concrete was divided into two portions—the coarse aggregate and the mortar—in order to ensure the fluidity of the concrete, as suggested by Okamura [5]. This was carried out by fixing the dosage of coarse aggregates in the concrete, and that of the sand in the mortar, by adjusting the quantities of water and superplasticizer (SP) [5]. In fact, there is a good correlation between the fresh-state behavior of the SCC and its mortar portion, regardless of the quantities of water and admixture used. In addition, tests on self-compacting mortar are easier to perform than those for SCC. However, the correlations could be affected by the use of supplementary cementitious materials, such as slag, natural pozzolan, and metakaolin. In recent years, there has

been an interest in using metakaolin as a partial replacement for cement in traditional or self-compacting concrete [6–9]. The use of supplementary cementitious materials, such as metakaolin (MK), as a partial substitution of cement in the production of SCC contributes to the reduction in energy consumption and greenhouse gas emissions, thus reducing the environmental impact of concrete [10]. MK is an ultra-fine pozzolana, composed mainly of silica and alumina. According to the French Standard, MK is produced by calcining kaolin clay within a specific temperature range, between 600 and 850 °C, depending on the chemical composition of the MK and the kaolinite classification degree [11]. Although there are some conflicting results reported in the literature, metakaolin generally tends to reduce the workability of concrete [12]. Because of the high specific surface area and the tendency of MK to agglomerate, both the plastic viscosity and yield stress of the concrete are reduced when both cement and MK are used [13]. However, the use of MK in limestone cement resulted in a reduction in the yield stress and an increase in the plastic viscosity [14] or an increase in both rheological properties [15]. The use of MK in SCC provides adequate flowability, passing ability, and viscosity by limiting the risks of bleeding and segregation [15]. Metakaolin has also been shown to improve the rheology and buildability of 3D-printed cement composite, as the static yield stress, dynamic yield stress, and viscosity increased with 10% MK compared with the control [16]. Few studies are available on the rheological behavior of limestone cement concrete with MK using a rheometer.

The main objective of this study was to investigate the validity of these correlations when using metakaolin. This paper presents some results of an extensive study on SCC [14], which aimed to extend the knowledge of the relationships between the fresh properties of SCC containing metakaolin (MK) and the spread of its related mortar portion. For this purpose, different percentages of a locally produced MK were used as partial substitutions of Portland limestone cement (PLC), in order to study the rheological behavior and the correlation between the various properties. The fresh properties of self-compacting mortars (SCMs) were obtained using the mini-cone spreading test and the V-funnel flow test. On the other hand, the properties of the SCC in the fresh state were evaluated according to the European Standard [17].

## 2. Experimental Design

### 2.1. Materials

In the present investigation, all SCC or SCM mixtures were formulated with a PLC cement, Type CEM II/A-L 42.5 R, containing 15% fine limestone, with a density equal to 3.03 g/cm$^3$ and a Blaine fineness of 4449 cm$^2$/g. The metakaolin (MK) used in this investigation was obtained from a local kaolin that was thermally activated at a temperature of 850 °C for 3 h. The MK obtained had a density of 2.6 g/cm$^3$ and a BET surface area of around 140,000 cm$^2$/g. The chemical compositions of the PLC cement and the MK used are reported elsewhere [14]. An ether polycarboxylate superplasticizer (SP), with a density of 1.07 g/cm$^3$, was used. Two coarse aggregates, with a maximum size of 3/8 mm and 8/15 mm and a density of 2.64 g/cm$^3$ and 2.66 g/cm$^3$, respectively, were also used. The fine aggregate used was an alluvial sand, with a density of 2.63 g/cm$^3$ and an absorption coefficient of 0.65%. The particle size distribution of the aggregates is presented in Table 1.

**Table 1.** Particle size distribution of aggregates.

| | Sieve Size (mm) | | | | | | | | | | | |
|---|---|---|---|---|---|---|---|---|---|---|---|---|
| | **16** | **12.5** | **10** | **8** | **6.3** | **5** | **2.5** | **1.25** | **0.63** | **0.315** | **0.16** | **0.08** |
| (Coarse aggregate A) 8–15 mm | 100 | 98 | 61 | 26 | 6 | 1 | 0 | – | – | – | – | – |
| (Coarse aggregate B) 3–8 mm | 100 | 100 | 100 | 98 | 77 | 50 | 5 | 1 | 1 | 1 | – | – |
| Fine aggregate | 100 | 100 | 100 | 100 | 100 | 100 | 99 | 98 | 91 | 51 | 9 | 2.55 |

*2.2. Methods*

The formulations of the SCC mixtures were based on the Okamura method [5]. The volume of fine aggregate in the mortar, the mass of added water, and the dosage of SP were selected based on the slump flow and V-funnel (Tv) tests [18]. For the selection of air content and coarse aggregates, the Okamura method was also employed. SCC is usually considered as a mortar matrix with coarse aggregates, and SCM could serve as a basis for the design of SCC. The workability of the SCC could be obtained from the spread and V-funnel tests of the SCM [19].

In fact, according to Domone [18], the spread of SC mortars between 280 and 340 mm leads to the spread in SCC between 550 and 850 mm as recommended by the European guidelines. However, the two viscosity classes of SCC provided by the V-funnel flow time of less than 8 s and 8 to 25 s correspond to the flow times of SCM of less than 3.6 s and between 3.6 and 13.1 s, respectively. Furthermore, Safiuddin [20] reported that a spread of SCM between 275 and 335 mm will generally produce an SCC with a spread of between 550 and 850 mm.

A mortar spread between 275 and 335 mm and a V-funnel flow time between 2 and 10 s were chosen as representative values for the characterization of SCM [20,21]. Bleeding and segregation were visually checked during the spreading test. After a preliminary investigation [22], a sand/mortar volume of 50%, an SP dosage of 1.1% by weight of cement, a water/cement (W/C) ratio of 0.42, and a cement dosage of 667 kg/m$^3$ were considered for the control SCM without MK. The compositions of the mortar (SCM) mixes are given in Table 2. The dosage of SP was adjusted based on the MK content as partial substitution of PLC cement [23].

**Table 2.** Composition of mortars.

| MK (%) | PLC | 5MK | 10MK | 15MK | 20MK | 25MK |
|---|---|---|---|---|---|---|
| SP (%) | 1.1 | 1.1 | 1.3 | 1.5 | 1.8 | 2.0 |
| W/P | 0.42 | 0.42 | 0.42 | 0.42 | 0.42 | 0.42 |
| Water (kg/m$^3$) | 280 | 280 | 280 | 280 | 280 | 280 |
| Powder (kg/m$^3$) | 667 | 667 | 667 | 667 | 667 | 667 |
| PLC (kg/m$^3$) | 667 | 634 | 600 | 567 | 534 | 500 |
| MK (kg/m$^3$) | 0 | 33 | 67 | 100 | 133 | 167 |
| Fine aggregate (kg/m$^3$) | 1316 | 1316 | 1316 | 1316 | 1316 | 1316 |

A total of six concrete mixes were employed with a MK content ranging from 0–25%, in increments of 5%, as partial cement (PLC) replacement for the formulation of the SCC. The water/binder (W/B) and gravel/sand ratios were kept constant at 0.42 and 1.0, respectively, with a binder content of 441 kg/m$^3$. The fine aggregate, coarse aggregate (3/8), and coarse aggregate (8/15) contents were 870 kg/m$^3$, 290 kg/m$^3$, and 580 kg/m$^3$, respectively. After fixing the sand volume and optimizing the water and SP dosages of the various SCMs, coarse aggregates for SCC mixes were selected from a range of 31 to 35% concrete volume to ensure an adequate flow and passing ability [21]. The air content was assumed to be equal to 1% for a maximum grain size (Dmax) of 20 mm [24]. The proportions of the SCCs formulated were in the typical ranges of the SCC constituents proposed by EFNARC [25] and RILEM [26]. The SCCs were characterized according to the guidelines [17,27] using the slump flow, V-funnel, L-box, sieve stability, and J-ring tests according to the standards for testing fresh concrete, EN 12350 Parts 8, 9, 10, 11, and 12, respectively. Table 3 summarizes the composition of the various concrete mixes.

**Table 3.** Composition of concrete mixtures.

| Mix Description | PLC | 5MK | 10MK | 15MK | 20MK | 25MK |
|---|---|---|---|---|---|---|
| W/P | 0.42 | 0.42 | 0.42 | 0.42 | 0.42 | 0.42 |
| Water (kg/m$^3$) | 185 | 185 | 185 | 185 | 185 | 185 |
| Powder (kg/m$^3$) | 441 | 441 | 441 | 441 | 441 | 441 |
| PLC (kg/m$^3$) | 441 | 419 | 397 | 375 | 353 | 331 |
| MK (kg/m$^3$) | 0 | 22 | 44 | 66 | 88 | 110 |
| Fine aggregate (kg/m$^3$) | 870 | 870 | 870 | 870 | 870 | 870 |
| Coarse aggregate 3/8 (kg/m$^3$) | 290 | 290 | 290 | 290 | 290 | 290 |
| Coarse aggregate 8/15 (kg/m$^3$) | 580 | 580 | 580 | 580 | 580 | 580 |
| Superplasticizer (%) | 1.1 | 1.1 | 1.3 | 1.5 | 1.8 | 2.0 |

## 3. Results and Discussion

A good relationship between the filling capacity of the SCCs, evaluated by the slump flow test and the fluidity of their related mortar portion, has been observed [14]. In this context, the form and nature of the relationships that may exist between the fresh state properties of SCCs containing MK and its mortar were investigated.

### 3.1. Filling Ability

Figures 1 and 2 and Table 4 show the effect of MK content on the rheological properties of the SCC. There was a slight increase in slump flow and little change in $T_{500}$ for the SCC containing MK (Figure 1). The increase in flow may be due to the increase in the SP dosage in the MK mixes. This means that the addition of MK to the SCC increased the filling capacity as expressed by the degree of filling (fluidity or spreading) and also increased the filling rate as expressed by the viscosity or the time $T_{500}$. However, this increase was marginal, as can be seen in Figure 1. Therefore, the effect of SCM constituents with a volume of 66% on the rheological properties of the SCC containing 33% aggregates (by volume) was further investigated by comparing the filling capacity, the spreading flow, and the resistance to the segregation of the SCC and the spreading of the related SCM.

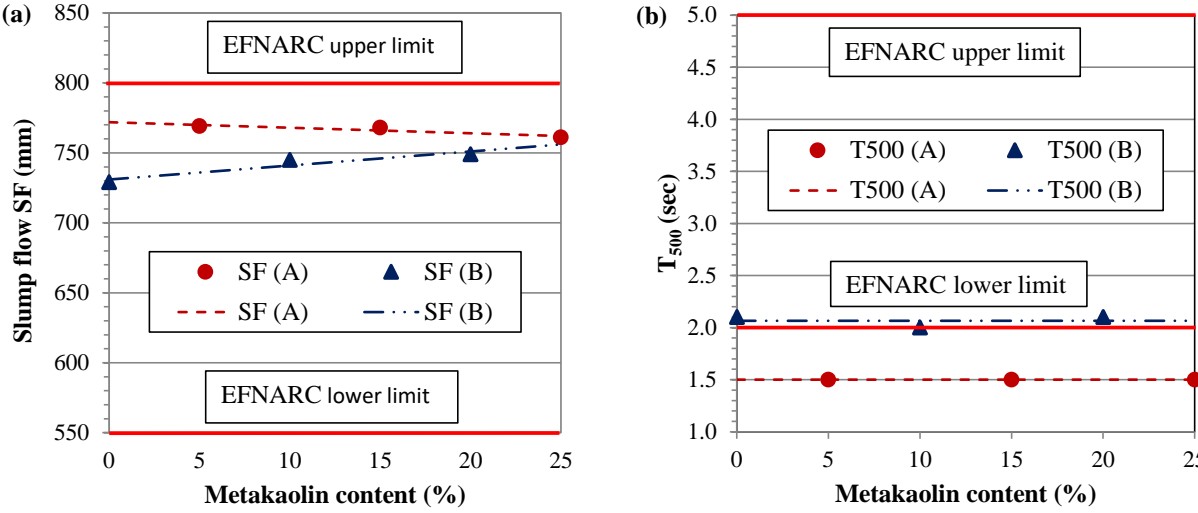

**Figure 1.** SCC filling ability tests. (**a**) SCC slump flow. (**b**) SCC $T_{500}$ slump flow time.

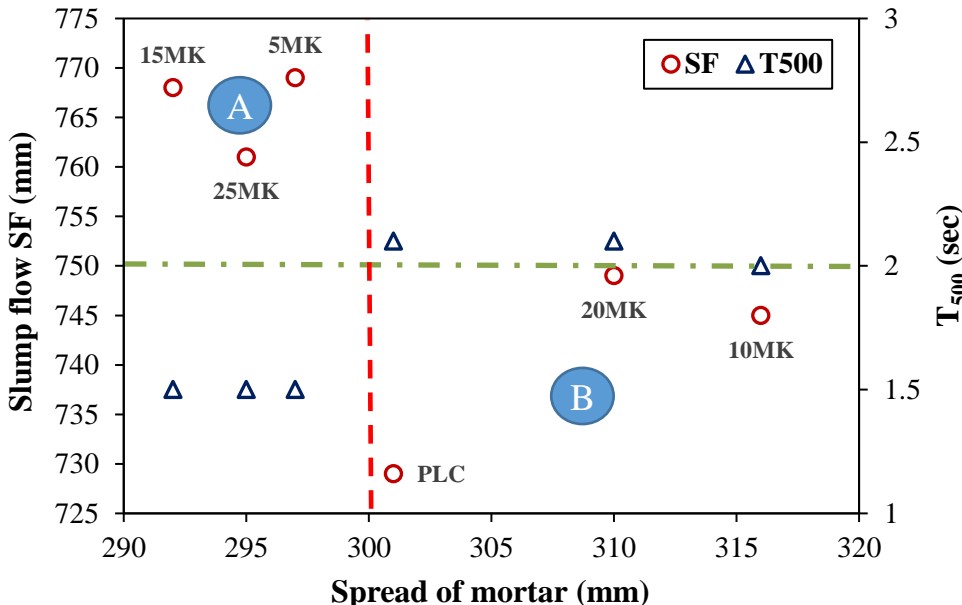

**Figure 2.** Filling ability tests of SCC vs. spread of mortar.

**Table 4.** Mortars and SCC categories. Filling ability of Category A and Category B SCCs.

| Category | A | | | B | | |
|---|---|---|---|---|---|---|
| MK (%) | 5MK | 15MK | 25MK | PLC | 10MK | 20MK |
| SP (%) | 1.1 | 1.5 | 2.0 | 1.1 | 1.3 | 1.8 |
| Spread (mm) | 297 | 292 | 295 | 301 | 316 | 310 |
| V-funnel (s) | 5.10 | 6.25 | 9.45 | 4.20 | 5.15 | 8.00 |
| Slump flow (mm) | 769 | 768 | 761 | 729 | 745 | 749 |
| $T_{500}$ (s) | 1.50 | 1.50 | 1.50 | 2.10 | 2.00 | 2.10 |

To obtain the relationships between mortar spread and the fresh properties of the SCC with the local metakaolin, the SCC mixtures were classified into two categories, according to the diameter of spread of the mortar [14]. Figure 2 suggests that the SCC mixtures can be classified into two categories of SCM: Category A, corresponding to a spread in the SCM of less than 300 mm, such as that of 5MK, 15MK, and 25MK, and Category B, with a spread in the SCM greater than or equal to 300 mm, such as that of PLC, 10MK, and 20MK. The first category had the SCC spreads above 750 mm and the flow time $T_{500}$ below 2 s, which correspond to the SCC with lower viscosity and a probable risk of segregation [27]. The opposite trend was seen in the second category (spreads of the SCCs ≤750 mm and flow time $T_{500}$ ≥2 s), which corresponds to the results of the required SCC [27].

It can be observed that the components of the SCMs played a major role in determining the fresh properties of their related SCC mixes. This was demonstrated by the good relationship between the SCMs and their related SCCs. This relationship depended, in particular, on the dosage of the SP and the MK content in the mix.

It can be seen that a gradual increase in MK in the SCC increased the fluidity of the SCC in Category B. The increase was 2.2% and 2.7% for the SCCs containing 10MK and 20MK, respectively. In the case of Category A, the fluidity decreased slightly from 769 mm to 768 mm and 761 mm for 5MK, 15MK, and 25MK, respectively (Figure 1a and Table 4). However, the viscosity of the SCCs, which is the resistance of a material to flow avoiding its internal frictions, was constant for each category. It was acceptable for Category B ($T_{500}$ ≥ 2 s) and lower for category A ($T_{500}$ < 2 s), as shown in Figure 1b and Table 4. The

flow rate of the SCCs with low viscosity was high at the beginning, then slowed down. On the other hand, the SCCs with high viscosity continued to flow slowly for a prolonged period, which contributed towards improving the suspension of the aggregates in the mix and, consequently, preventing the segregation and keeping the homogeneity of the mix.

Usually, the SP had mainly an effect on the spreading, whereas the water content strongly affected both the fluidity and viscosity of the mixtures. From Table 4, it is clearly seen that there was an irregular deformability at the level of the SCM, and this could be due to the effect of the SP.

The high fluidity and low viscosity of the SCCs in Category A (Figure 2) may be due to the higher water demand in the presence of MK. Moreover, a high viscosity is necessary for the segregation resistance of SCC, but should not be excessively high so that the coarse aggregate cannot pass through the space between the rebars [28]. Therefore and in order to increase the segregation resistance of Category A SCCs, it was necessary to increase the viscosity, by reducing the W/B ratio, or by incorporating a viscosity-enhancing agent (VEA), or by using aggregates with a maximum diameter of 10 mm. According to Yahia et al. [29], adequate resistance to segregation is obtained by reducing the W/C ratio, increasing the cohesion of the paste, adding finer particles, or by using a viscosity agent. Furthermore, Chai [24] reported that a VEA can be used if the W/B is greater than 0.37 for an SCC with a Dmax equal to 20 mm, and no VEA is required if the W/B is between 0.40 and 0.50 and the Dmax is around 10 mm.

According to Barbhuiya [30] and Khayat [31], the viscosity of concrete can be improved by either increasing the binder content or by incorporating a VEA. The binder content can be achieved by lowering the W/B in order to maintain adequate cohesive friction between the mortar and coarse aggregate and ensure the uniform flow of the SCC through the reinforcing bars. The use of VEA allows the reduction in the volume of coarse aggregates and reduces the risk of blockage, which is particularly useful in mixes containing a moderate content of supplementary cementitious materials and fillers. Furthermore, the use of VEA in the SCC is beneficial when using discontinuous, angular, plate, and elongated coarse aggregates and a lower content of cementitious materials [28].

Finally, the targeted average value of the SCM (spread of $305 \pm 10$ mm) resulted in an SCC spread between 729 and 769 mm and flow times $T_{500}$ between 1.50 and 2.10 s. In order to avoid the probability of having too low a viscosity or the risk of the possible segregation of the SCC mixtures with a spread above 750 mm, the workability and the ability to fill Category A concretes formulated in this way must be checked using the SCC workability tests so that the necessary adjustments can be made as follows:

1. Either adjusting the water content of Category B SCCs in such a way that there will be a spread ≤750 mm and a flow time $T_{500}$ ≥2 s. However, the W/B ratio will not be constant for all the mixtures;
2. Adding a VEA to Category A SCCs, which belong to the spreading class SF3 (760 to 850 mm), whose resistance to segregation is more difficult to control [25].

According to Safiuddin [28], it is recommended to use a VEA in SCCs when the mixtures are too fluid and present a risk of segregation, which should be improved without changing the primary proportions of the concrete. However, the Japanese approach used in this work was developed for concretes without VEA and was extended to concretes with VEA.

The spread value in mortar that was found to be acceptable (i.e. spread ≥ 300 mm), was suggested by Chai [21] to produce a successful SCC. However, other researchers have found a successful SCC with spread values in SC mortars below 300 mm. It has been reported that the Japanese experience suggested that values between 250 and 280 mm for mini-cone spreading and 6 and 10 s for V-funnel flow time can produce a successful SCC [21]. A target spread of 240 to 260 mm and a V-funnel flow time of 7 to 11 s were suggested by EFNARC [27]. Target values for spread and V-funnel flow times, equal to 245 mm and 10 s, respectively, have also been suggested [32].

Therefore, in order to produce a successful SCC, the choice of mortar spread value should aim for an SCC spread target value. According to Chai [24], a spread value between 600 and 700 mm and between 650 and 750 mm will be sufficient for SCCs with a Dmax of 10 and 20 mm, respectively. Below the lower limit, concrete may have insufficient fluidity to pass through and around obstacles, and above the upper limit, segregation is likely to occur [33]. Moreover, 90% of the cases analyzed by Domone [34], the SCC formulated around the world, had spreads between 600 and 750 mm. Furthermore, according to the AFGC [1], the target spread value is generally between 600 and 750 mm.

### 3.2. V-Funnel Flow Time of SCC vs. Spread of Mortar

The measurement of the V-funnel flow time (Tv) is considered as an alternative to that of the flow time $T_{500}$ [35]. The results of $T_{500}$ and Tv were used to evaluate the viscosity and fluidity of the SCCs. The fluidity of SCMs with MK was reduced and the viscosity was increased for cement substitution by 5% of MK at a constant SP dosage and a constant W/B ratio [36]. According to Hassan et al. [22], the increase in $T_{500}$ or Tv indicates an increase in the viscosity or a decrease in the fluidity of the mixture. Figure 3 shows clearly that the viscosity of 5MK, evaluated by the time Tv, decreased contrary to $T_{500}$ and the fluidity of the SCC increased instead of being reduced (Figure 1). This may be due to the fine limestone of the PLC cement used, which reduced both the water content of the mix and the interfacial transition zone (ITZ), by contributing to the lubrication of the coarse aggregates and consequently improving the fluidity of the SCC at low levels of MK.

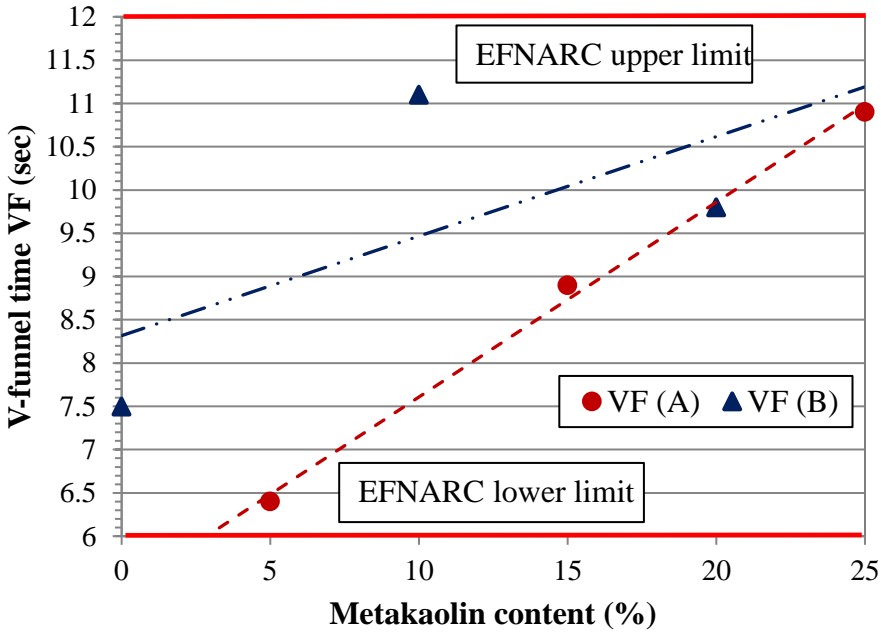

**Figure 3.** V-funnel flow time for the various SCC mixtures.

According to Figure 3 and as the dosage of SP increased, the viscosity increased, as shown by the increase of the flow time (Tv) compared to the control concrete for mixes with MK content above 10%. This was in agreement with results obtained elsewhere [23]. According to Hassan et al. [22], the increase in $T_{500}$ or Tv indicates an increase in the viscosity or a decrease in the fluidity of the mixture. The clay nature of MK contributes to the increase in viscosity and, consequently, decreases the fluidity and the risk of segregation of SCC.

However, it can be seen clearly that the mixtures of 15MK and 25MK showed a successive increase in their flow time relative to their successive decrease in their spreading. On the other hand, the opposite trend was observed for mixtures 10MK and 20MK. It can also be noted that there was a good relationship between the spread of the SCM and

the filling capacity of the related SCC measured by the V-funnel test (Figure 4) and that the level of cement substitution with MK still affected the SCCs according to their mortar spreading diameter. Increasing the content of MK decreased the Tv of SCCs for slump flow of the SCM of more than 300 mm (Category B) and increased at the same time the Tv of SCCs for the SCM with a slump flow less than 300 mm (Category A). This indicates that the substitution of cement by MK did not change the correlations established between the SCC and its related SCM.

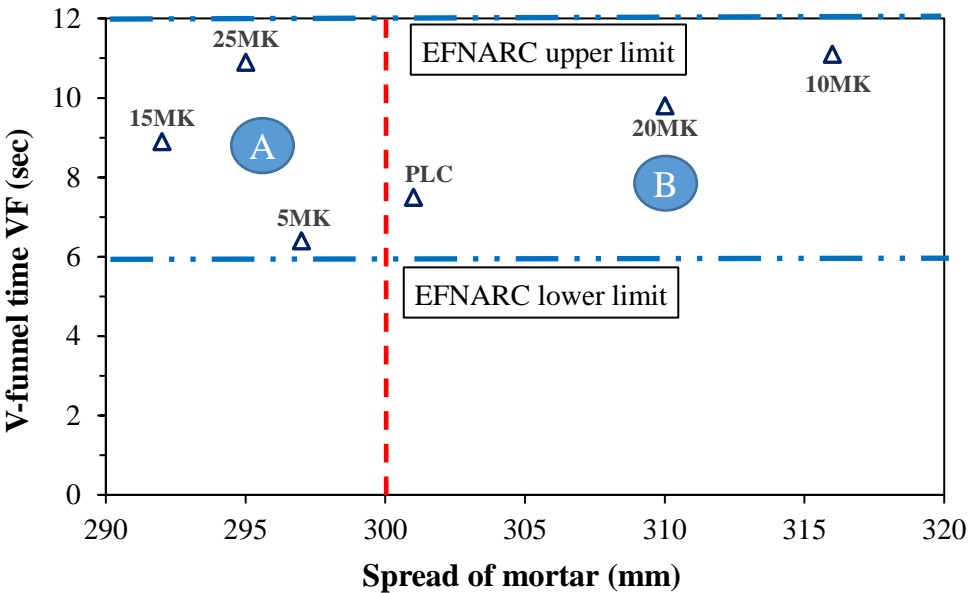

**Figure 4.** V-funnel flow time of SCC versus spread of mortar.

For a particle size Dmax of 20 mm, all the formulated SCC mixes were within the viscosity range (Tv = 6–12 s) and a flow time of 10 s, which is considered more suitable for SCCs as recommended by EFNARC [27]. According to Domone et al. [33], if the flow time is higher than 10 s, the concrete is either too viscous for satisfactory handling and placement or is unstable as the aggregate particles can block the flow of the SCC. SCC mixes with 10% and 25% of MK gave a flow time of 11.10 s and 10.90 s, respectively, and a spread flow of 745 and 761 mm, respectively, and hence, it can be concluded that the SCC with 10MK (Category B) was too viscous for satisfactory handling and placement. On the other hand, the SCC with 25MK (Category A) was unstable, so that the aggregates particles blocked the flow of the SCC.

### 3.3. L-Box Test of SCC vs. Spread of Mortar

The L-box test is used to measure the filling and passing capacity of the SCC through confined areas [37–39] with no segregation or blockage. However, in order to obtain adequate passing capacity, the coarse aggregate content should be less than the maximum amount recommended (i.e., 35% of the concrete volume) [28]. On the other hand, increasing the cement content from 450 kg/m$^3$ to 500 kg/m$^3$, the passing capacity of the mixtures will be greatly improved [22].

The ability to flow or pass, in narrow openings and areas of high reinforcement density, is assessed by measuring the filling rate, expressed by the ratio of the height of the concrete in the vertical part of the box and the height of the concrete at the end of the horizontal part of the box (H2/H1). The concrete flow time values $t_{200}$ and $t_{400}$ (the times when the concrete reaches the distances 200 mm and 400 mm from the horizontal arm, respectively) are used to evaluate the speed deformation of the SCC and give an indication of the ease of concrete flow. These measured parameters showed similar trends to those of the filling capacity measured by the slump flow test (Figure 1).

According to Figure 5, it can be seen that the partial substitution of PLC cement by MK increased the value of filling rate (H2/H1 ≥ 0.8) for passing through three bars, which is suitable for areas with narrower and denser reinforcement and decreased the flow times t200 and t400. The mixtures of SCCs with MK had a higher passage capacity than that of the control concrete. Similar results have been found by other researchers, indicating that the addition of supplementary cementitious materials increases the passing capacity of the SCC [19]. Moreover, MK has been shown to increase the viscosity of the mixture and to contribute to the improvement of particles' suspension in the mixtures, thus leading to greater passing capacity and greater resistance to segregation. However, the rate of incorporation of MK greatly affects the flowability of the SCCs according to the spreading diameter of their mortars. The increase in MK content in the SCCs increased as the value of the filling rate increased according to the category type. The values of Category A (0.84; 0.88; 0.88) were slightly higher than the SCCs of Category B (0.80; 0.83; 0.86). On the other hand, for the flow times t200 and t400, the values of Category A increased and the values of Category B decreased systematically as the content of MK increased. This indicates that the substitution of cement with MK did not change the correlations established between the mortar and its SCC. It can be concluded that the results of the L-box test exhibited similar trends to those reported previously [13]. Furthermore, there was a good relationship between the spread of the SCM and the flowability of the related SCC measured by the L-box test, as illustrated by Figure 5.

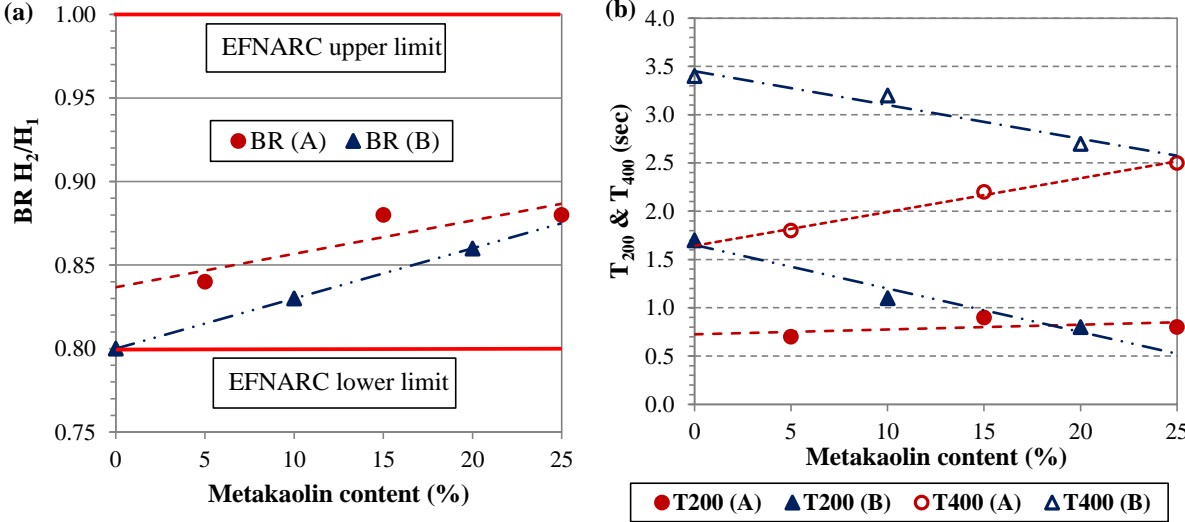

**Figure 5.** SCC L-box tests. (**a**) SCC blocking ratio. (**b**) SCC $T_{200}$ and $T_{400}$.

Sonebi et al. [38] reported that the filling rate represented by the H2/H1 ratio is influenced by three parameters, the water dosage, the SP dosage, and the volume of gravel in the mixture. The increase in the first two parameters leads to an increase in the filling rate. However, the increase in the volume of gravel leads to a reduction in the filling rate, thus increasing the risk of the blockage of the coarse aggregates behind the steel bars of the L-shaped box. Furthermore, it can be noted that the increase in the filling rate of the SCC is due to the increase in the first two parameters: the W/B ratio, which is greater than 0.37 for an SCC with a Dmax equal to 20 mm, and the SP dosage, which increases with increasing levels of MK. Nevertheless, this increase in the filling rate still depends on the spreading diameter of the mortar.

In addition, SCCs (Category A) with a higher filling rate and low $t_{200}$ and $t_{400}$ values had the ability to flow in the presence of obstacles compared to those of Category B. Moreover, the incorporation of MK reduced the filling rate of Category A due to the associated increase in viscosity [39]. SCC flowability was 0.84, 0.88, and 0.88 for the 5MK, 15MK, and 25MK mixes, respectively, which is in the range of (0.8–1.0) suggested by

EFNARC for a maximum particle size Dmax of 20 mm [27], but slightly out of the range proposed according to the Swedish experience for good passing ability [26]. This was the case for SCCs of Category B, where the flowability was 0.80, 0.83, and 0.86 for the PLC, 10MK, and 20MK mixes, respectively. On the other hand, the absence of VEA increased the flow of the SCCs by increasing the filling rate as those observed for the SCCs of Category A with high fluidity, which may be due to their instability.

*3.4. J-Ring Difference Height of SCC vs. Spread of Mortar*

The J-ring flow test is used to determine the passing ability of SCC through confined areas with high reinforcement density [27,40,41]. It can be used as an alternative to the L-box test. However, the results are not directly comparable [37]. The degree of filling of the ring flow spread with class PJ2 was greater than the value of 10 mm suggested by EFNARC [27] for all the mixtures containing MK, as shown in Figure 6. These SCC mixtures with MK can be chosen for structures with widely spaced reinforcements [41]. The 5MK mix had a high blocking rate, which may be due to its instability, where the aggregate particles blocked the flow. Furthermore, it can be indicated that the 5MK mix had a similar tendency to that of the flow time Tv, which had a very low viscosity. The viscosity of the mortar must be high to resist the separation of coarse aggregates. However, it should not be excessively high so that the coarse aggregates cannot pass through the space between the rebars.

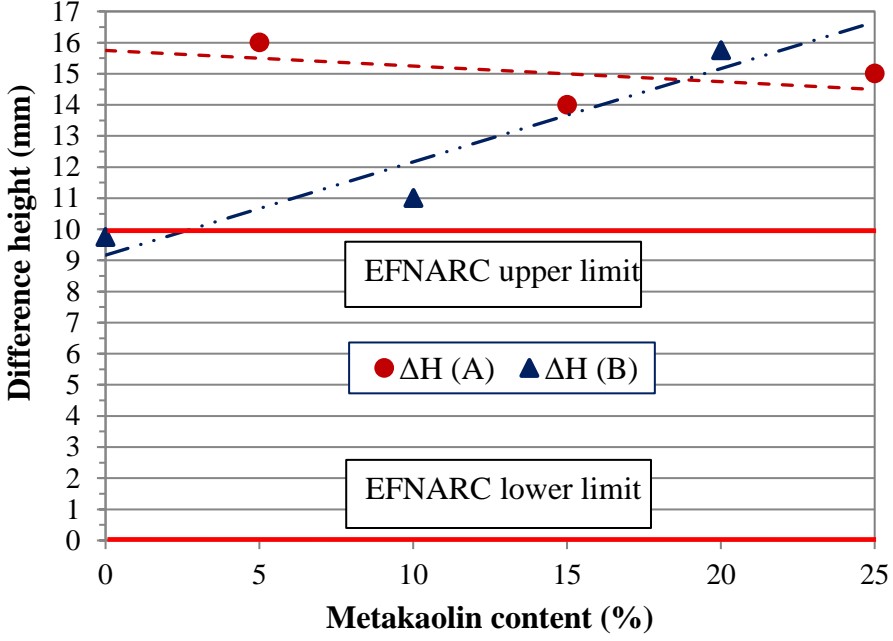

**Figure 6.** SCC J-ring height difference.

Figure 6 indicates that the filling rates of the ring spreading of the SCCs had similar trends to their mortar spreading diameter. It was noticed that, as the MK content in the SCC mixtures increased, there was an increase in the degree of filling of Category B mixtures, whereas a decrease was observed in those of Category A.

Figure 7 shows the flow with and without the J-ring for various MK contents. The rate of substitution of cement by MK influenced the spread of the SCCs, and this was related to their mortar spreading diameter (i.e. 300 mm). In the case of the SCCs of Category A, the flow spread was higher than that of Category B. However, the flow time was lower for Category A. The flow times with and without the J-ring diverged from each other due to the decrease in the flow capacity with an increased risk of blockage. Furthermore, the difference between the flow spread with and without the J-ring of SCCs containing MK increased between 25 and 50 mm as the MK content increased. This clearly indicates a

remarkable blockage, which makes the concrete more suitable for normal applications with unreinforced or lightly reinforced concrete sections.

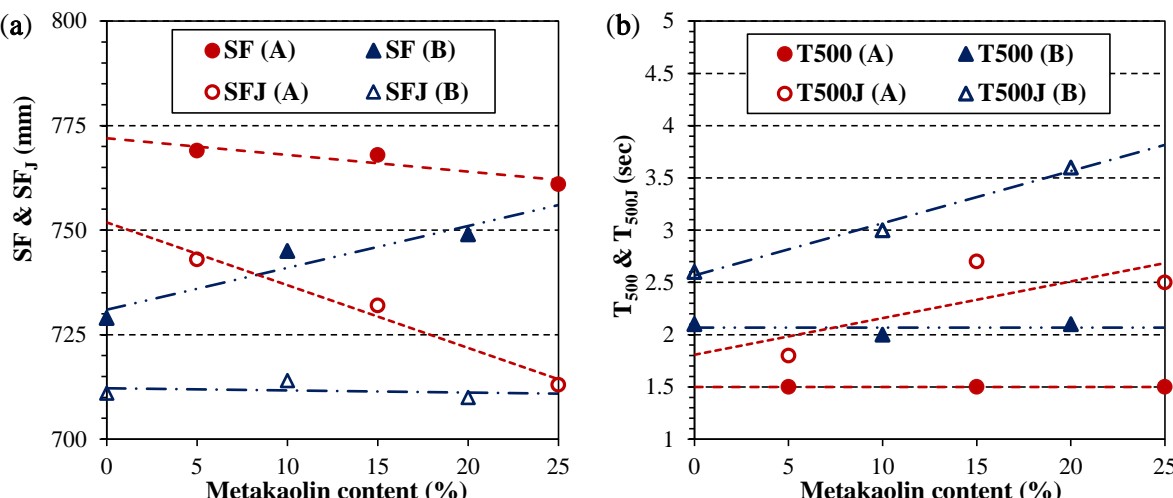

**Figure 7.** SCC filling ability for (**a**) slump flow (SF) and (**b**) $T_{500}$ with and without J-ring for different metakaolin contents.

*3.5. Segregation Resistance Test of SCC vs. Spread of Mortar*

The segregation of SCC occurs mainly as a separation between coarse aggregates and the mortar portion [42]. The resistance to sieving has been used to evaluate the segregation of coarse aggregates of SCC mixes. Figure 8 illustrates the segregation resistance of the SCCs versus the flow spread of their mortars. It can be seen that all SCC mixtures with and without MK presented a homogeneous and stable concrete. All the SCCs had a segregation rate between 5 and 15%, which is classified as Class SR2 (15%). This limit of 5 to 15% is recommended for optimal resistance to segregation [41]. It can be observed from this figure that the SCCs of Category B were more stable than those of Category A.

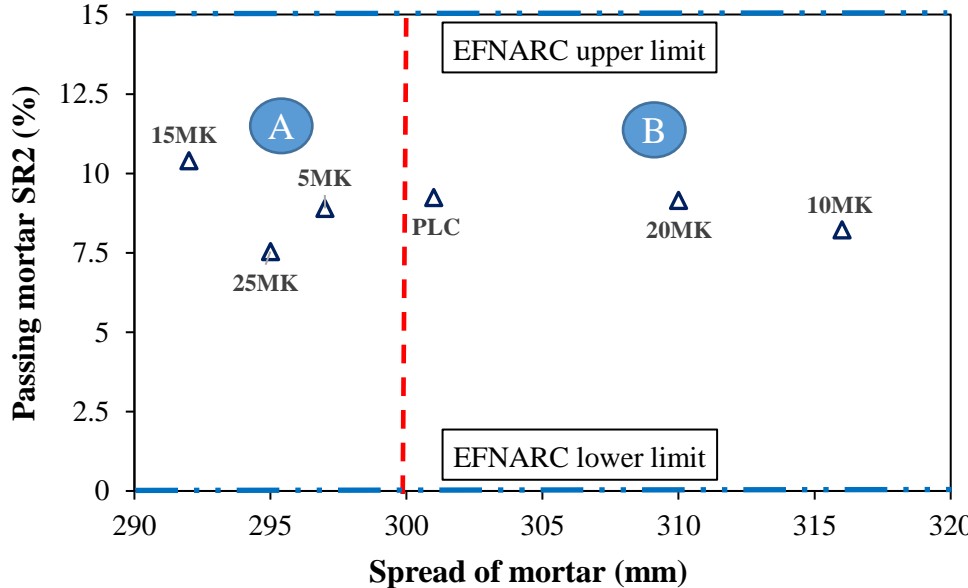

**Figure 8.** Segregation resistance of SCC versus spread of mortar.

According to the EN 206-9 standard [17], the stability of the SCC is an important parameter to be taken into consideration, in the case of higher spread and lower viscosity classes. This is the case of the SCCs of Category A (i.e., spread of SCM < 300 mm), where

the spreads conformed to the highest Class SF3 (760 to 850 mm) and the viscosity to the lowest class (i.e., VS1 < 2 s). To mitigate this problem, a VEA should be used. On the other hand, the segregation of coarse aggregates is strongly affected by the viscosity of the mixture, where its increase reduces the probability of segregation. The fluidity of concrete could be maintained by adding SP, whereas the stability and reduction of segregation and bleeding are maintained by modifying the VEA and powder content [43]. According to Saifuddin [28], the optimum flow capacity and resistance to segregation can be obtained by adjusting the fluidity and viscosity of the concrete through an appropriate combination of cement and a supplementary cementitious material, by limiting the W/B ratio and adding an appropriate dosage of SP and, possibly, adding an appropriate dosage of VEA.

## 4. Conclusions

Based on this experimental investigation on the formulation of SCC through its SCM, the following conclusions can be given:

✓ A good relationship exists between the spreading of SCM and the fresh properties of the related SCC. The choice of spreading of SCM by more than 300 mm using PLC, 10MK, and 20MK led to the desired properties of the SCC.

✓ The content of MK as cement substitution does not change the relationship between the SCM and its related SCC properties such as slump flow spread, V-funnel time, L-box filling rate, J-ring height difference and segregation resistance.

✓ The choice of spreading value of the SCM can be used to obtain the SCC fresh properties with a Dmax equal to 20 mm. These include a spread value of the SCC between 600 mm and 750 mm, a flow time (Tv) of 10 s, a filing rate value between 0.80 and 0.85, and a Pj value less than 10 mm.

✓ The use of VEA for an SCC with a higher spread (Class SF) and low viscosity (Class VS1) is needed to have a good resistance to segregation.

✓ Further studies are recommended to check the validity of the correlations stated in this investigation using other supplementary cementitious materials such as slag, natural pozzolan, and calcined clay. Multiple tests for each rheological property are required to check the statistical significance of the tests and the correlations.

**Author Contributions:** Conceptualization, S.K. and B.M.; methodology, S.K., A.B. and E.-H.K.; software, A.B.; validation, S.K., J.K. and B.M.; formal analysis, A.B.; investigation, A.B.; resources, E.-H.K.; data curation, A.B.; writing—original draft preparation, A.B. and B.M.; writing—review and editing, S.K. and J.K.; visualization, S.K. and J.K.; supervision, S.K.; project administration, S.K.; funding acquisition, S.K., E.-H.K. and J.K. All authors have read and agreed to the published version of the manuscript.

**Funding:** The APC was partially funded by L2GMC laboratory, CY Cergy-Paris University.

**Data Availability Statement:** Data are available upon request.

**Conflicts of Interest:** The authors declare no conflict of interest.

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
