# Peer review of "Relationships between Mortar Spread and the Fresh Properties of SCC Containing Local Metakaolin"

_infrastructures, doi:10.3390/infrastructures8100137_

Round 1
Reviewer 1 Report
There are many aspects that need to be checked and improved.
A check of English is expected.
References not in English should be avoided.
PhD thesis references should be avoided.
The Abstract needs to be improved.
The novelty of the paper needs to be better enhanced.
Lines 16-18 need to be better explained.
The introduction needs to be improved. More recent references need to be added to the paper.
References not in English should be avoided.
The experimental program section needs to be improved.
A Table with all the mix design samples in order of MK % increase needs to be added to better enhance the differences among the different mixes and understand the obtained fluctuation in all the results. What about the use of VEA?
The procedure used to prepare the six mixes need to be better explained.
Lines 73-79 need to be better explained.
Line 83 needs to be better explained (i.e., test stage).
Line 102: the meaning of (G/S) can be better elucidated.
The results and Discussion section needs to be extensively improved (lines 106-370).
The differences in the 6 mix designs need to be better correlated to the obtained results.
Lines 107-110 are not clear and need to be better explained.
Figure 1 is not clear and needs to be better explained in the text. More numbers in the figure can be added to the y-axes to better understand the fluctuation of values. The cause of this behavior needs to be better explained in the text. It is not clear if results were obtained in one test, or if they correspond to a mean value. The deviation standard needs to be added.
Figure 2 needs to be better described in the test. It is not clear why a value of 300 mm was selected to divide the materials into two classes. The division in two classes is not clear.
Lines 115-119 need to be better explained.
Table 2 needs to be better introduced and explained in the paper. Lines 121-124 need to be better explained. Is it 0MK or PLC?
All results (lines 106-370) need to be better correlated to the differences among the different mix designs.
Conclusions are not clear and need to be improved also according to previous observations.
A check of English is expected.
Author Response
Infrastructures
Paper: Relationships Between Mortar Spread and the Fresh Properties of SCC with Local Metakaolin
Dear Editor,
The authors would like to thank the editor and the reviewers for their valuable comments to improve the quality of the paper.
All the comments and suggestions have been dealt with thoroughly. Here is an account of how it was dealt with the comments one by one. The modifications are shown in the annotated file in red color.
Reply to reviewers:
Reviewer 1:
- A check of English is expected.
The English language has been checked thoroughly by a native English speaker and modified accordingly. All the modifications are shown in red color.
- References not in English should be avoided.
The references in French are eight (08) PhD theses and (01) one paper and they have been avoided as suggested in comment 3 by the same reviewer.
- PhD thesis references should be avoided.
The eight (08) PhD thesis references in French have deleted form the list of references as suggested. Only four (04) theses in English remain.
- The Abstract needs to be improved.
The following sentence has been added to the abstract: (Spreading of mortar is used to estimate SCC rheological properties but the use of supplementary cementitious materials such as metakaolin could affect the accuracy of the estimation).
In addition, the last sentence has been rewritten as follows: (The choice of the mortar flow should be done according to the needed rheological properties of the SCC and a viscosity enhancing agent (VEA) could be used if the mix is too fluid and present an instability).
- The novelty of the paper needs to be better enhanced.
The novelty of the paper is now better explained in the last paragraph of the introduction sentence.
(However, the correlation could be affected by the use of supplementary cementitious materials such as slag, natural pozzolan and metakaolin. The main objective of this paper is to investigate the validity of these correlations when using metakaolin).
- Lines 16-18 need to be better explained.
Lines 16 to 18 have been rewritten as follows: (The choice of the mortar flow should be done according to the needed rheological properties of the SCC and a viscosity enhancing agent (VEA) could be used if the mix is too fluid and present an instability).
- The introduction needs to be improved. More recent references need to be added to the paper.
Five (05) references have been added to the introduction section (references 2, 3, 10, 11 and 12).
- The experimental program section needs to be improved.
A Table with all the mix design samples in order of MK % increase needs to be added to better enhance the differences among the different mixes and understand the obtained fluctuation in all the results. What about the use of VEA?
The procedure used to prepare the six mixes need to be better explained.
A table (Table 2) has been added to give the mix details of the mortar and another table (Table 3) has been added to show the SCC mixes. VEA were not used as the Japanese general method was developed for mixes without VEA. The procedure to prepare the mixes is now better explained.
- Lines 73-79 need to be better explained.
This part of section 2.2 has been rewritten as follows:
(The formulation of the SCC mixtures was based on the general Okamura method. The volume of sand in the mortar and the dosages (in mass) of water and SP of binder were selected based on the slump flow and the V-funnel tests [15] For the selection of air contents and coarse aggregates, the Okamura method was employed. SCC is usually considered as a mortar matrix with coarse aggregate and the SCM could serve as a basis for the design of SCC for which the workability could be obtained from the spread test and the v-funnel test of SCM [16].).
- Line 83 needs to be better explained (i.e., test stage).
This has been modified to read (preliminary investigation).
- Line 102: the meaning of (G/S) can be better elucidated.
G/S is now explained in the text as Gravel/Sand ratio.
- The results and Discussion section needs to be extensively improved (lines 106-370).
The results discussion section has been improved where appropriate throughout the text.
- The differences in the 6 mix designs need to be better correlated to the obtained results.
All the figures given concern the various correlations obtained.
- Figure 1 is not clear and needs to be better explained in the text. More numbers in the figure can be added to the y-axes to better understand the fluctuation of values. The cause of this behavior needs to be better explained in the text. It is not clear if results were obtained in one test, or if they correspond to a mean value. The deviation standard needs to be added.
Figure 1 explanation is now improved.
- Figure 2 needs to be better described in the test. It is not clear why a value of 300 mm was selected to divide the materials into two classes. The division in two classes is not clear.
Figure 2 is now better explained in the text.
- Lines 107-110 are not clear and need to be better explained.
Lines 115-119 need to be better explained.
The paragraph in section 3.1 has been rewritten as follows: (There a slight increase in slump flow and little change in T500 for SCC containing MK. The increase in of flow may be due to the increase in SP dosage in MK mixes. This means that the addition of MK to SCC increases the filling capacity as expressed by the degree of filling (fluidity or spreading) and also increases the filling rate as expressed by the viscosity or the time T500). However, this increase is not regular as can be seen in Fig.1. Therefore, the effect of the constituents of SCM with a volume of 66% on the rheological properties of SCC containing 33% aggregates (by volume) will be further investigated by comparing on one side the filling capacity, the spreading flow and the resistance to segregation of SCC and on the other side the spreading of the related SCC).
- Table 2 needs to be better introduced and explained in the paper. Lines 121-124 need to be better explained. Is it 0MK or PLC?
Table 2 has now been introduced in the beginning of section 3.1 as follows: (Table 4 and figures 1 and 2 summarize the effect of MK content on the rheological properties of SCC).
It is PLC. This has now been corrected.
- All results (lines 106-370) need to be better correlated to the differences among the different mix designs.
The results discussion has been improved where appropriate throughout the section.
- Conclusions are not clear and need to be improved also according to previous observations.
The conclusion section has been improved.
The second conclusion has been rewritten as follows: (The content of MK as cement substitution does not change the relationship between the SCM and its related SCC properties such as slump flow spread, v-funnel time, L-Box filling rate, J-ring height difference and segregation resistance).
The last conclusion was rewritten as follows: (The use of VEA for SCC with higher spread (class SF) and low viscosity (class VS1) is needed to have a good resistance to segregation).
- Comments on the Quality of English Language: A check of English is expected.
The English language has been thoroughly checked by a native English speaker and modified accordingly.

Reviewer 2 Report
This study investigated the relationship between the fresh properties of self-compacting concrete (SCC) containing local metakaolin (MK) and the flowability of its mortar. The results showed strong correlations between mortar flowability and the properties of fresh self-compacting concrete.
Please find my main remarks below:
1. The article should clearly include the final compositions of the tested mixtures, for example, in tabular form
2. Authors should rethink the form of data presentation in the figures 1, 3, 5, 6, 7. A solid line connecting single results suggests that this is the course of the trend. This is not appropriate if the studied characteristic was determined on a single sample. If the studied feature was determined in multiple samples, this should be emphasized and presented, for example, in the form of a box plot or by specifying the standard deviation. Should the authors really want to present this as a relationship then a trend line (function) should be derived, along with the determination of the correlation or determination coefficient.
3. I think the article should be revisited for typos and inconsistent formatting
4. The conclusions are unsatisfactory. They should be expanded. Conclusions should summarize the entire article and indicate the overall conclusion of the study. The implications of the study's findings for practice and potential real-world applications are worth noting. It is worth pointing out areas that require further research.
Author Response
Infrastructures
Paper: Relationships Between Mortar Spread and the Fresh Properties of SCC with Local Metakaolin
Dear Editor,
The authors would like to thank the editor and the reviewers for their valuable comments to improve the quality of the paper.
All the comments and suggestions have been dealt with thoroughly. Here is an account of how it was dealt with the comments one by one. The modifications are shown in the annotated file in red color.
Reply to reviewers:
Reviewer 2:
Comments and Suggestions for Authors
This study investigated the relationship between the fresh properties of self-compacting concrete (SCC) containing local metakaolin (MK) and the flowability of its mortar. The results showed strong correlations between mortar flowability and the properties of fresh self-compacting concrete.
Please find my main remarks below:
1. The article should clearly include the final compositions of the tested mixtures, for example, in tabular form
Two (02) tables have been added for the mixes of mortar and concrete (Tables 2 and 3).
- Authors should rethink the form of data presentation in the figures 1, 3, 5, 6, 7. A solid line connecting single results suggests that this is the course of the trend. This is not appropriate if the studied characteristic was determined on a single sample. If the studied feature was determined in multiple samples, this should be emphasized and presented, for example, in the form of a box plot or by specifying the standard deviation. Should the authors really want to present this as a relationship then a trend line (function) should be derived, along with the determination of the correlation or determination coefficient.
All the figures have been redrawn as suggested. The rheological properties are determined on one sample and hence no standard deviation is given.
3. I think the article should be revisited for typos and inconsistent formatting
The English language has been checked and improved where necessary by a native English speaker.
4. The conclusions are unsatisfactory. They should be expanded. Conclusions should summarize the entire article and indicate the overall conclusion of the study. The implications of the study's findings for practice and potential real-world applications are worth noting. It is worth pointing out areas that require further research.
The conclusion section has been improved.
The second conclusion has been rewritten as follows: (The content of MK as cement substitution does not change the relationship between the SCM and its related SCC properties such as slump flow spread, v-funnel time, L-Box filling rate, J-ring height difference and segregation resistance).
The last conclusion was rewritten as follows: (The use of VEA for SCC with higher spread (class SF) and low viscosity (class VS1) is needed to have a good resistance to segregation).
In addition, the following suggestion for further research was added (Further studies are needed to check the validity of these correlation with other supplementary cementitious materials such as slag, natural pozzolan and calcined clay and with replicating the rheological tests to show the statistical significance).

Reviewer 3 Report
The authors have conducted a study concerning the fresh properties of Self-Compacting Concrete (SCC) and its correlation to Supplementary Cementitious Material (SCM) spread performance across varying Metakaolin (MK) substitution proportions. Overall, the methodology of the study is correct, and the discussion provided is richly detailed. Nevertheless, it would be beneficial if the authors could articulate more clearly the significance of this study. The direct relationship between mortar spread characteristics and SCC fresh properties is well-established in the field. Without a more specific explanation regarding how MK usage may affect this correlation, the reviewer questions the necessity for reconfirmation.
Based on the manuscript, there is no replication of any batch of mixture. Is that correct? Though SCC and mortar can be relatively stable, the reviewer thinks the data may not be reliable enough. Please consider replicating the mixing and measurement to include the variety of the material and mixing process.
Apart from the clarity regarding the study's motivation and method, there are a few aspects of the presentation that could benefit from further revisions.
1. The size distribution of the aggregates can be presented in a better form. Though experienced researchers can understand Table 1, it would be better to present the data in curves rather than unitless numbers.
2. The mixed design can be better presented in section 2.1 instead of 2.2. A table can help organize and compare the amount of material. Mix design should be part of the materials rather than methods.
3. It is ok to refer to references when describing the method and discussing the results, but too much will make the description wordy and overwhelm the author's work. Also, the current introduction does not have sufficient background context, especially in the details. The reviewer suggests reorganizing the references and summarizing them in the introduction section rather than spreading them all across the sections, which can help make the later sections more concise.
4. Please recheck the format of the captions etc.
5. The reviewer suggests redrawing most of the figures which have reference lines. Many of the reference lines seem inaccurate (not horizontal or not matching the value). Please consider using professional plotting software instead of MS Office.
Author Response
Infrastructures
Paper: Relationships Between Mortar Spread and the Fresh Properties of SCC with Local Metakaolin
Dear Editor,
The authors would like to thank the editor and the reviewers for their valuable comments to improve the quality of the paper.
All the comments and suggestions have been dealt with thoroughly. Here is an account of how it was dealt with the comments one by one. The modifications are shown in the annotated file in red color.
Reply to reviewers:
Reviewer 3:
Comments and Suggestions for Authors
The authors have conducted a study concerning the fresh properties of Self-Compacting Concrete (SCC) and its correlation to Supplementary Cementitious Material (SCM) spread performance across varying Metakaolin (MK) substitution proportions. Overall, the methodology of the study is correct, and the discussion provided is richly detailed.
- Nevertheless, it would be beneficial if the authors could articulate more clearly the significance of this study. The direct relationship between mortar spread characteristics and SCC fresh properties is well-established in the field. Without a more specific explanation regarding how MK usage may affect this correlation, the reviewer questions the necessity for reconfirmation.
The novelty of the paper is better explained now in the introduction section. The idea is to see whether the correlations can also be used for the case of metakaolin.
- Based on the manuscript, there is no replication of any batch of mixture. Is that correct? Though SCC and mortar can be relatively stable, the reviewer thinks the data may not be reliable enough. Please consider replicating the mixing and measurement to include the variety of the material and mixing process.
The rheological properties are measured on one sample and there is no replication.
Apart from the clarity regarding the study's motivation and method, there are a few aspects of the presentation that could benefit from further revisions.
- The size distribution of the aggregates can be presented in a better form. Though experienced researchers can understand Table 1, it would be better to present the data in curves rather than unitless numbers.
The aggregates size distribution is resented in tabular form to avoid using a figure that was published elsewhere by the same authors (Reference 13).
- The mixed design can be better presented in section 2.1 instead of 2.2. A table can help organize and compare the amount of material. Mix design should be part of the materials rather than methods.
We agree with reviewer and section 2.1 contains only the materials used. The mix design is in section 2.2.
- It is ok to refer to references when describing the method and discussing the results, but too much will make the description wordy and overwhelm the author's work. Also, the current introduction does not have sufficient background context, especially in the details. The reviewer suggests reorganizing the references and summarizing them in the introduction section rather than spreading them all across the sections, which can help make the later sections more concise.
Some references were deleted and other were added. Thos deleted are the theses in French and five references were added (they are now references 2, 3, 10, 11 and 12).
- Please recheck the format of the captions etc.
The format of the captions has been checked
- The reviewer suggests redrawing most of the figures which have reference lines. Many of the reference lines seem inaccurate (not horizontal or not matching the value). Please consider using professional plotting software instead of MS Office.
All figures have been redrawn as suggested.

Reviewer 4 Report
A manuscript titled “ Relationships Between Mortar Spread and the Fresh Properties of SCC with Local Metakaolin.” was submitted to the journal “Infrastructures”, manuscript highlighted the importance of metakaolin in SCC and mortar.
Why mortar not concrete? This does not make sense. SCC is a concrete, not a mortar.
Here are my notes and comments on the paper.
Abstract
Too short, it should be improved well, and it should contain results at the end of the paragraph.
Introduction
This section is not enough to prove that metakaolin is necessary to be used in the SCC. There are many studies that are missed to discuss on them, such as “Pressure exerted on formwork by self-compacting concrete at early ages: A review”, authors should understand how the rheology of SCC is worked and how metakaolin will help.
This section should be related studies direct to metakaolin and SCC studies which has a great workability for the congested reinforcement and those spots that do not reach concrete easily.
Experimental Program
If authors used over 4.6 mm then that is concrete not mortar, this is confusing, authors should clearly understand what is mortar and what is concrete.
I cannot see any novel idea or test preparation in this section. Please provide more test and experimental studies if it is available.
Results and Discussion
Figure 1 why the varies the slump if there is 5% or 10% of the metakaolin in the test? Please update.
Figure 2 it is an image taken from somewhere, if this image is not belong to the authors please cite and take a permission.
Figure 6 why the SCC J-ring are tested in differents, authors should tell all details, is there any purpose?for those test you have a limited you cannot increase or decrease them. Please explain.
This section is too short and I cannot see very robust discussion for the results. It should have some mechanical strength at hard concrete stage or tests for shrinkage.
Conclusion
This is too short section, authors should increase of bullet by having more results and outcomes, and should also see what is left to be done in future.
Reference
Authors should include all related papers to this study in terms of SCC rheology and Metakaolin properties
The English language must be improved and make it flow for the reader.
Author Response
Infrastructures
Paper: Relationships Between Mortar Spread and the Fresh Properties of SCC with Local Metakaolin
Dear Editor,
The authors would like to thank the editor and the reviewers for their valuable comments to improve the quality of the paper.
All the comments and suggestions have been dealt with thoroughly. Here is an account of how it was dealt with the comments one by one. The modifications are shown in the annotated file in red color.
Reply to reviewers:
Reviewer 4:
Comments and Suggestions for Authors
A manuscript titled “ Relationships Between Mortar Spread and the Fresh Properties of SCC with Local Metakaolin.” was submitted to the journal “Infrastructures”, manuscript highlighted the importance of metakaolin in SCC and mortar.
- Why mortar not concrete? This does not make sense. SCC is a concrete, not a mortar. Here are my notes and comments on the paper.
The tests are conducted on mortar and correlations with the rheological properties of concrete are established.
- Abstract: Too short, it should be improved well, and it should contain results at the end of the paragraph.
The abstract has been improved as also suggested by reviewer 1, comment 4. The following sentence has been added to the abstract: (Spreading of mortar is used to estimate SCC rheological properties but the use of supplementary cementitious materials such as metakaolin could affect the accuracy of the estimation).
In addition, the last sentence has been rewritten as follows: (The choice of the mortar flow should be done according to the needed rheological properties of the SCC and a viscosity enhancing agent (VEA) could be used if the mix is too fluid and present an instability).
- Introduction: This section is not enough to prove that metakaolin is necessary to be used in the SCC. There are many studies that are missed to discuss on them, such as “Pressure exerted on formwork by self-compacting concrete at early ages: A review”, authors should understand how the rheology of SCC is worked and how metakaolin will help.This section should be related studies direct to metakaolin and SCC studies which has a great workability for the congested reinforcement and those spots that do not reach concrete easily.
The introduction section has been improved. The suggested reference though not directly related has been added (reference 3). Another three references have also been added. The following paragraph was also added for better explanation of the use of MK.
(However, the correlations could be affected by the use of supplementary cementitious materials such as slag, natural pozzolan and metakaolin. In recent years, there has been an interest in using metakaolin as partial replacement of cement in traditional or self-compacting concrete [6-9]. The use of supplementary cementitious materials such as metakaolin (MK) as partial substitution of cement in the production of SCC contribute to the reduction of cement cost production by reducing the energy consumption and reducing the greenhouse gas emissions thus reducing the environmental impact and help preserve natural resources by reducing the industrial-by-product amount dumped in the landfill [10]. MK is an ultra-fine pozzolana, composed mainly of silica and alumina and is produced by calcining kaolin clay within a specific temperature range between 600 to 850 °C depending on chemical composition of MK and the kaolinite classification degree, according to the French standard [11]. The use of MK in SCC provides adequate flowability, passing ability and viscosity by limiting the risks of bleeding and segregation [12]. The main objective of this study is to investigate the validity of these correlations when using metakaolin).
- Experimental Program: If authors used over 4.6 mm then that is concrete not mortar, this is confusing, authors should clearly understand what is mortar and what is concrete. I cannot see any novel idea or test preparation in this section. Please provide more test and experimental studies if it is available.
The authors disagree with the reviewer. Aggregates of less than 4.6 mm are fine aggregates and when mixed with cement, they result in mortar. Concrete should contain coarse aggregates of more than 5 mm size. Unfortunately, no more tests are available.
- Results and Discussion: Figure 1 why the varies the slump if there is 5% or 10% of the metakaolin in the test? Please update.
The reason of slump flow variation with the slump content is explained in the text and it is to do with the fineness of MK.
- Figure 2 it is an image taken from somewhere, if this image is not belong to the authors please cite and take a permission.
Figure 2 was mistakenly taken as image but it is the author own figure. It has been redrawn.
- Figure 6 why the SCC J-ring are tested in different, authors should tell all details, is there any purpose? for those test you have a limited you cannot increase or decrease them. Please explain. This section is too short and I cannot see very robust discussion for the results. It should have some mechanical strength at hard concrete stage or tests for shrinkage.
The J ring test was carried out according to its European standard EN 12350-12 and EN 206-9. On the other hand, the SCC filling ability tests with and without J-Ring were added. This study concerns only the fresh state properties of the concrete
- Conclusion: This is too short section, authors should increase of bullet by having more results and outcomes, and should also see what is left to be done in future.
The conclusion section has been improved.
The second conclusion has been rewritten as follows: (The content of MK as cement substitution does not change the relationship between the SCM and its related SCC properties such as slump flow spread, v-funnel time, L-Box filling rate, J-ring height difference and segregation resistance).
The last conclusion was rewritten as follows: (The use of VEA for SCC with higher spread (class SF) and low viscosity (class VS1) is needed to have a good resistance to segregation).
In addition, the following suggestion for further research was added (Further studies are needed to check the validity of these correlation with other supplementary cementitious materials such as slag, natural pozzolan and calcined clay and with replicating the rheological tests to show the statistical significance).
- Reference: Authors should include all related papers to this study in terms of SCC rheology and Metakaolin properties
Some references were deleted and other were added. Thos deleted are the theses in French and five (05) references were added to the introduction section (they are now references 2, 3, 10, 11 and 12).
- Comments on the Quality of English Language: The English language must be improved and make it flow for the reader.
The English language has been thoroughly checked by a native English speaker and modified accordingly.

Round 2
Reviewer 1 Report
Figure 1 seems to have different results with reference to the original version of the paper. This aspect can be better elucidated. What about the original fluctuations? What about the standard deviation?
A check of typing errors is still expected.
Author Response
INFRASTRUCTURES
Paper: Relationships Between Mortar Spread and the Fresh Properties of SCC with Local Metakaolin
Reviewer 1-R2
The authors would like to thank the editor and the reviewer for their comments. Here is an account of how it was dealt with the two comments and the modifications in the paper are in red color.
- Comments and Suggestions for Authors
Figure 1 seems to have different results with reference to the original version of the paper. This aspect can be better elucidated. What about the original fluctuations? What about the standard deviation?
Authors’ Reply
No, the results are the same but, in this version, instead of grouping all the results on one curve, the category A and category B are separated and plotted on different curves and hence the fluctuations do not appear. Only one rheological test is performed for each property and no standard deviation can be obtained. It has been recommended for future work to perform multiple tests for each property for replication purposes.
The following sentence has been added to the last conclusion at the end of the paper: (Multiple tests for each rheological property are required to check the statistical significance of the tests and the correlations).
- Comments on the Quality of English Language
A check of typing errors is still expected.
Authors’ Reply
The authors have checked the whole manuscript and they spotted some minor spelling mistakes that were corrected and are in red color (lines 47, 49, 86, 110, 146, 189, 196, 207, 208, 231, 293 and 316).

Reviewer 3 Report
The quality of the paper is improved and the main concern on the motivation has been answered, but still can be improved. For example, Fg.1, two fonts can be found and some reference lines seems inaccurate (inclined). The reviewer suggest further proofread and redraw the figures with professional plotting software.
The other concern is the data may not be reliable as there is no replicant. Please provide some discussion on this limitation and the possible impact on your result.
Author Response
INFRASTRUCTURES
Paper: Relationships Between Mortar Spread and the Fresh Properties of SCC with Local Metakaolin
The authors would like to thank the editor and the reviewer for their comments. Here is an account of how it was dealt with the two comment and the modifications in the paper are in red color.
Reviewer 3-R2
Comments and Suggestions for Authors
- The quality of the paper is improved and the main concern on the motivation has been answered, but still can be improved. For example, Fg.1, two fonts can be found and some reference lines seems inaccurate (inclined). The reviewer suggest further proofread and redraw the figures with professional plotting software.
Authors’ Reply
Figure 1 has been redrawn for better clarity and the legend expanded and better explained.
There are no two fonts but may be it is just the coincidence with the lines of the graph parallel to the y-axis.
- The other concern is the data may not be reliable as there is no replicant. Please provide some discussion on this limitation and the possible impact on your result.
Authors’ Reply
Usually, the rheological tests are performed only once as these properties will change with time. Nevertheless, to show the limitation of this study, a suggestion for further research has been added to the last conclusion at the end of the paper as follows: (Multiple tests for each rheological property are required to check the statistical significance of the tests and the correlations).

Reviewer 4 Report
There is still some editing and arrangement that can be made in the table and improve the introduction of the manuscript.
It can be revised again and some proofreading for the manuscript.
Author Response
INFRASTRUCTURES
Paper: Relationships Between Mortar Spread and the Fresh Properties of SCC with Local Metakaolin
The authors would like to thank the editor and the reviewer for their comments. Here is an account of how it was dealt with the two comment and the modifications in the paper are in red color.
Reviewer 4-R2
- Comments and Suggestions for Authors
There is still some editing and arrangement that can be made in the table and improve the introduction of the manuscript.
Authors’ reply:
Tables 1, 2, 3 and 4 have been rearranged.
The introduction of the manuscript has been improved by adding the following paragraph (lines 54 to 65) and three references (now 12, 13 and 16): (Although, there are some conflicting results reported in the literature, metakaolin tends generally to negatively affect the workability of concrete [12]. Because of the high specific surface area and the tendency of MK to agglomerate, both plastic viscosity and yield stress are reduced for binary blend cements (cement and metakaolin) [13]. However, the use of MK in limestone cement resulted in reduction in yield stress and an increase in plastic viscosity [14] or an increase in both rheological properties [15]. The use of MK in SCC provides adequate flowability, passing ability and viscosity by limiting the risks of bleeding and segregation [15]. Metakaolin has also been shown to improve the rheology and buildability of 3D printed cement composite as static yield stress, dynamic yield stress, and viscosity increased with 10% MK compared with control [16]. Few studies are available on the rheological behavior of limestone cement concrete with MK using a rheometer).
- Comments on the Quality of English Language
It can be revised again and some proofreading for the manuscript.
Authors’ reply:
The authors have checked the whole manuscript and they spotted some minor spelling mistakes that were corrected and are in red color (lines 47, 49, 86, 110, 146, 189, 196, 207, 208, 231, 293 and 316).
